# FedPS: Federated data Preprocessing via aggregated Statistics

## Abstract

Data preprocessing is a crucial step in machine learning that significantly influences model accuracy and performance. In Federated Learning (FL), where multiple entities collaboratively train a model using decentralized data, the importance of preprocessing is often overlooked. This is particularly true in Non-IID settings, where clients hold heterogeneous datasets, requiring aggregated parameter estimates to perform consistent data preprocessing. In this paper, we introduce FedPS, a comprehensive suite of tools for federated data preprocessing. FedPS leverages aggregated statistics, data sketching, and federated machine learning models to address the challenges posed by distributed and diverse datasets in FL. Additionally, we resolve key numerical issues in power transforms by improving numerical stability through log-space computations and constrained optimization. Our proposed Federated Power Transform algorithm, based on Brent's method, achieves superlinear convergence. Experimental results demonstrate the impact of effective data preprocessing in federated learning, highlighting FedPS as a versatile and robust solution compared to existing frameworks. The implementation of FedPS is open-sourced.

## 1 Introduction

Data preprocessing (García et al., 2016) plays a crucial role in data mining and machine learning, ensuring that raw data—often fraught with missing values and inconsistencies—can be refined into a form suitable for model training. Proper preprocessing not only enhances the robustness of the training process but also significantly boosts model accuracy. However, in Federated Learning (Kairouz et al., 2021), where data is stored locally across multiple clients and models are trained collaboratively, centralizing the data for preprocessing is not feasible due to privacy concerns and decentralized storage. This presents a unique challenge, as traditional preprocessing steps, which are typically applied before data distribution in centralized simulations, cannot be directly implemented in a federated setting.

In federated environments, preprocessing requires the estimation of statistics from decentralized data, which must then be aggregated on the server. While some statistics like min, max, sum, mean, and variance are straightforward to compute with minimal communication overhead, others—such as quantiles and frequent items—pose significant challenges due to computational and communication constraints.

This paper introduces FedPS, a comprehensive suite of tools designed to tackle these preprocessing challenges in FL. Leveraging data sketching techniques (Cormode & Yi, 2020), which efficiently summarize large datasets while retaining critical information, FedPS enables the computation of both simple and complex statistics in a distributed manner. The concept of mergeability (Agarwal et al., 2013) further supports the federated learning setting by allowing sketches from different clients to be combined efficiently. We also extend several key algorithms, such as Bayesian Linear Regression (Tipping, 2001), to both horizontal and vertical federated settings.

Additionally, our paper conducts an in-depth analysis of existing data preprocessors in the widely-used *Scikit-learn* (Pedregosa et al., 2011) library, implementing federated versions while maintaining the flexibility and functionality of the original modules. Unlike other federated learning libraries, such as *FATE* (Liu et al., 2021) and *SecretFlow* (The SecretFlow Authors, 2022), which offer lim-

ited preprocessing options, our approach provides a full range of preprocessors with customizable parameters, making it more versatile and powerful.

In tackling the numerical challenges of the power transform, previously identified by Marchand et al. (2022) but not fully resolved in practice, we propose a effective solution. By performing calculations in log space, we also introduce constrained optimization to improve numerical stability. Furthermore, we develop a Federated Power Transform algorithm using Brent's method (Brent, 2013), which achieves superlinear convergence, outperforming previous approaches that relied on slower exponential search methods (Marchand et al., 2022), offering only a linear convergence rate.

Our main contributions are as follows:

- Implementation of a comprehensive suite of federated data preprocessing tools, utilizing aggregated statistics, data sketching.

- Addressing the numerical issues identified in power transform through log space computations and constrained optimization.

- Extending Bayesian Linear Regression to both Horizontal and Vertical federated learning setting. And proposing a federated power transform algorithm with a superlinear convergence rate.

- Open-sourcing the implementation of FedPS.

The remainder of the paper is structured as follows: Section 2 outlines our motivation. Section 3 provides a review of existing techniques, laying the foundation for the technical aspects of federated preprocessing discussed in Section 4. Our solution to the power transform's numerical issues and the corresponding federated algorithm are detailed in Section 5. Section 6 presents experimental results, followed by related work in Section 7. Finally, we conclude the paper in Section 8.

## 2 MOTIVATION

**Boosting Model Performance.** Data preprocessing plays a pivotal role in enhancing the accuracy and performance of machine learning models. While much attention has been directed towards optimizing federated training algorithms, the significance of preprocessing data in a distributed manner cannot be overlooked. In our experiments, we aim to shed light on this aspect by contrasting the test accuracy achieved using raw data against that obtained using preprocessed data. Through this comparison, we seek to demonstrate the impact of federated data preprocessing on model performance, highlighting its potential to significantly boost accuracy.

**Necessity of Federated Computation.** In federated learning, data is distributed across multiple clients, preprocessing steps need to be adapted to this decentralized nature. While a decentralized strategy, with each client independently conducting preprocessing locally, may be suitable for scenarios with independent and identically distributed (IID) data, it encounters difficulties in non-IID scenarios. In such cases, clients may possess varied data distributions, such as label distribution skew, where each client exclusively holds one type of labeled data. To illustrate the significance of federated data preprocessing, consider a scenario where two parties collaboratively train a horizontally federated classification model. The initial data is linearly separable when pooled (see Figure 1(a)). However, when each party's data has distinct target categories (e.g., party A's data labeled 0 and party B's data labeled 1), applying local scaling for zero mean and unit variance results in non-linearly separable data (see Figure 1(b)). Consequently, federated data preprocessing becomes indispensable.

**Robust (Federated) Power Transform.** Power transform is widely utilized across various domains, including genomic studies (Zwiener et al., 2014) and geochemical data analysis (Howarth & Earle, 1979). Previous research (Marchand et al., 2022) has highlighted numerical challenges associated with power transform, yet adequate solutions remain elusive. Their solution relies on exponential search, resulting in linear convergence rates. In contrast, our approach involves a comprehensive theoretical analysis of the underlying numerical instabilities and presents an effective solution. Furthermore, we extend our methodology to federated settings and employ Brent's method, known for its superlinear convergence property, thereby offering a more robust and efficient approach.

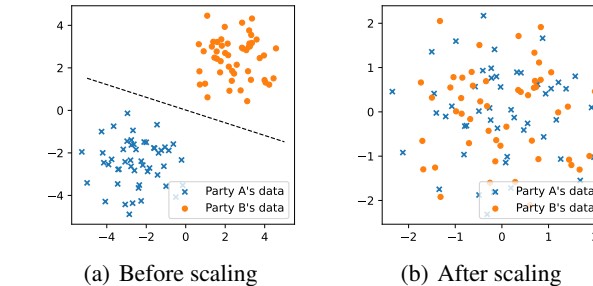

(a) Before scaling  (b) After scaling

Figure 1: The impact of feature scaling on label-skewed data if each client independently conducting preprocessing locally.

## 3 PRELIMINARIES

In this section, we begin with an overview of common data preprocessing steps. Subsequently, we delve into the background of federated learning. We also conduct a review of relevant aggregated statistics employed in our implementation and introduce the background of power transform.

### 3.1 DATA PREPROCESSING

Data preprocessing involves a diverse set of methods for preparing data. Common steps encompass feature scaling, encoding, discretization, missing value imputation, and various transformation methods tailored to specific scenarios. Our focus in this paper is on the preprocessors presented in *Scikit-learn*[1] (Pedregosa et al., 2011). The preprocessing workflow includes setting up the preprocessor with user-defined parameters, estimating the preprocessing parameters by calling the `fit` method, and finally, using the `transform` method to yield transformed data using the learned parameters. A review of the preprocessors in *Scikit-learn* is deferred to Appendix A.

### 3.2 FEDERATED LEARNING

Federated learning is a setting where data is decentralized, and immediate results are exchanged for aggregation to achieve a common learning objective. Two typical data partition axes are horizontal (example-partitioned) and vertical (feature-partitioned). In the horizontal setting, each client has the same feature space, while in the vertical setting, they share the same ID. As data preprocessing is often applied to each feature, most of the federated preprocessors presented in this paper are designed for the horizontal setting.

### 3.3 AGGREGATING STATISTICS

In federated learning, individual clients generate their local statistics and send to the server. Subsequently, these statistics are collected and aggregated by a central server, enabling queries and obtaining global estimations. A straightforward example is Min/Max, where each client computes its local minimum or maximum and transmits it to the server to obtain the global value. In the following paragraphs, we will provide a brief overview of other statistics used in our implementations.

**Sum, Mean, Variance.** These statistics involve maintaining counts. Sum has one counter. Mean has two counters: $c$ for the sum of data and $n$ for the number of examples. The mean value is calculated as $c/n$. Variance introduces another counter, $s$, representing the sum of squared data. It's computed as $s/n - (c/n)^2$ (Cormode & Yi, 2020). When merging counter-based statistics, simply add the corresponding counters.

**Quantiles.** Quantiles represent ordered statistics, associating values with specific ranks in sorted data. For instance, the median corresponds to quantile 0.5. Obtaining exact quantiles requires

---

[1]https://scikit-learn.org/stable/modules/preprocessing.html

maintaining information proportionate to the full data size, leading to many quantile sketches being approximate. Two common types of errors associated with approximate quantile sketches are additive error (Karnin et al., 2016) and multiplicative error (Cormode et al., 2023).

**Set Union, Frequent Items.** The union operation is executed by the server after receiving local sets from all clients. This process primarily utilizes hash tables. The frequent items sketch (Anderson et al., 2017), also known as heavy hitters, aims to track the frequency of each item in the set.

*DataSketches* (The DataSketches Authors, 2023) is an open-source library that provides fast streaming algorithms for big data. It includes sketches for quantiles, frequent items, and more. We leverage these sketches from this library in our implementation of federated data preprocessing.

### 3.4 POWER TRANSFORM

The power transform is a data transformation technique employed to make data more Gaussian distribution-like. Two well-known transformations for this purpose are Box-Cox (BC) (Box & Cox, 1964) and Yeo-Johnson (YJ) (Yeo, 2000). It's essential to note that Box-Cox requires input data to be strictly positive (i.e., $x > 0$), while Yeo-Johnson extends its applicability to both positive and negative data. The transformation functions for both methods are continuous and defined as follows, with visualizations provided in Figure 2.

$$\psi_{\text{BC}}(\lambda, x) = \begin{cases} (x^\lambda - 1)/\lambda & \text{if } \lambda \neq 0, \\ \ln x & \text{if } \lambda = 0. \end{cases} \tag{1}$$

$$\psi_{\text{YJ}}(\lambda, x) = \begin{cases} [(x + 1)^\lambda - 1]/\lambda & \text{if } \lambda \neq 0, x \geq 0, \\ \ln(x + 1) & \text{if } \lambda = 0, x \geq 0, \\ [-(-x + 1)^{2-\lambda} - 1]/(2 - \lambda) & \text{if } \lambda \neq 2, x < 0, \\ -\ln(-x + 1) & \text{if } \lambda = 2, x < 0. \end{cases} \tag{2}$$

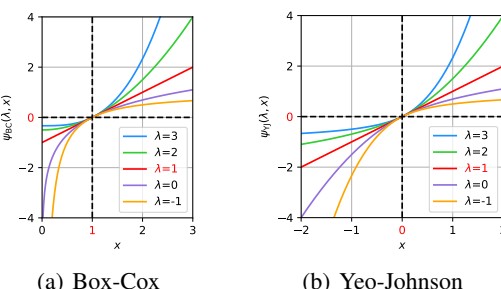

(a) Box-Cox      (b) Yeo-Johnson

Figure 2: Box-Cox and Yeo-Johnson transformation functions.

The power parameter $\lambda$ is estimated by minimizing the negative log-likelihood function, as defined in Equation 3 and 4. Notably, the negative log-likelihood functions for both Box-Cox and Yeo-Johnson transformations have been proven to be strictly convex (Kouider & Chen, 1995; Marchand et al., 2022), indicating that the function exhibits a unique global minimum.

$$-\ln \mathcal{L}_{\text{BC}}(\lambda, x) = (1 - \lambda) \sum_i^n \ln x_i + \frac{n}{2} \ln \sigma^2_{\psi_{\text{BC}}(\lambda, x)} \tag{3}$$

$$-\ln \mathcal{L}_{\text{YJ}}(\lambda, x) = (1 - \lambda) \sum_i^n \text{sgn}(x_i) \ln(|x_i| + 1) + \frac{n}{2} \ln \sigma^2_{\psi_{\text{YJ}}(\lambda, x)} \tag{4}$$

In the implementation within *SciPy* (Virtanen et al., 2020), the one-dimensional minimization for the power transform utilizes Brent's method (Brent, 2013). This algorithm efficiently evaluates the target function at a small number of points and converges superlinearly.

Table 1: Preprocessors and associated statistics.

| Categories | Preprocessors | Formulation | Associated Statistics |
|---|---|---|---|
| Scaling | MaxAbsScaler | $x/|x|_{\mathbf{max}}$ | Max |
| | MinMaxScaler | $(x - x_{\mathbf{min}})/(x_{\mathbf{max}} - x_{\mathbf{min}})$ | Min, Max |
| | StandardScaler | $(x - \mu)/\sigma$ | Mean, Variance |
| | RobustScaler | $(x - q_{0.5})/(q_{0.75} - q_{0.25})$ | Quantiles |
| | Normalizer | $x/\|x\|$ | Sum, Max |
| Encoding | FeatureHasher | $\mathrm{hash}(x)$ | – |
| | OneHotEncoder | $\text{one-hot}(x)$ | Set Union, Frequent items |
| | OrdinalEncoder | $\mathrm{ordinal}(x)$ | Set Union, Frequent items |
| | TargetEncoder | $\lambda(n_i)\frac{n_{iY}}{n_i} + (1 - \lambda(n_i))\frac{n_Y}{n}$ | Set Union, Mean, Variance |
| | LabelBinarizer | $\text{one-hot}(y)$ | Set Union |
| | MultiLabelBinarizer | $\text{multi-hot}(y)$ | Set Union |
| | LabelEncoder | $\mathrm{ordinal}(y)$ | Set Union |
| Transformation | FunctionTransformer | $f(x)$ | –* |
| | PowerTransformer | $\psi(\lambda, x)$ | Sum, Mean, Variance, Mix, Max |
| | QuantileTransformer | $\mathrm{CDF}(x), \Phi^{-1}(\mathrm{CDF}(x))$ | Quantiles |
| | SplineTransformer | $\text{B-spline}(x)$ | Min, Max, Quantiles |
| Discretization | Binarizer | 1 if $x > T$ else 0 | – |
| | KBinsDiscretizer | $j$ if $T_j \leq x < T_{j+1}$ | Min, Max, Quantiles, Mean |
| Imputation | SimpleImputer | $\mathrm{mean}(x), \mathrm{median}(x), \text{most-freq}(x)$ | Mean, Quantiles, Frequent items |
| | IterativeImputer | $\mathrm{RegressionModel}(x)$ | Sum |
| | KNNImputer | $\mathrm{mean}(k\text{-nearest neighbors of } x)$ | Horizontal: Min, Mean; Vertical: Sum |

*Only if the transformation function is stateless

# 4 FEDERATED DATA PREPROCESSING

The overview of federated data preprocessing steps is illustrated in Figure 3. Initially, each client generates its local statistics and transmits it to the server. Subsequently, the server performs the merging step on clients' summaries, and the server queries the merged summaries to obtain the necessary preprocessing parameters. Finally, these parameters are communicated back to the clients for the execution of data preprocessing.

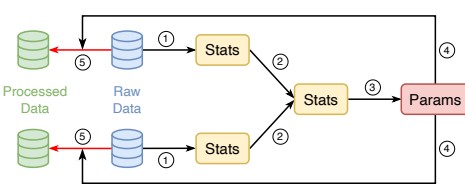

Figure 3: An overview of federated data preprocessing steps.

We categorize data preprocessors into distinct groups (scaling, encoding, transformation, imputation) to improve clarity regarding their functionalities. Within each category, we summarize the formulation and required statistics for each module, as outlined in Table 1. Additionally, Table 4 provides a comprehensive overview of the statistics associated with preprocessors and their communication cost. As most statistics are directly related to the functionality of each module , such as MinMaxScaler requiring computation of global minimum and maximum values, we focus on explaining the most significant ones.

**Scaling.** In RobustScaler, we utilize a quantile sketch to obtain the necessary quantiles. A unique scenario arises with Normalizer, particularly in vertical federated learning settings, where computing the global norm of each sample is necessary. For $l_1$ or $l_2$ norms, the computation involves obtaining the global sum of $|x|$ or $x^2$ for each sample (then taking the square root for $l_2$ norm); for the max norm, simply compute the global maximum of $|x|$.

**Encoding.** In federated learning, it's crucial for all clients to agree on a unified encoding scheme to ensure consistent encoding of the same categorical value into the same numeric value. Thus, encoding modules need to compute set unions, except for FeatureHasher, which relies on hash functions. Additionally, we utilize frequent items sketch in OneHotEncoder and OrdinalEncoder to ignore infrequent items. For TargetEncoder, the global mean is required, along with variance for determining the smoothing parameter.

**Transformation.** Regarding FunctionTransformer, if the user-provided function is stateless (i.e., requires no parameter estimation from the data), then the aggregation isn't necessary. However, in the case of PowerTransformer, aggregation is required for evaluating the negative log-likelihood function (Equation 3 and 4) multiple times, necessitating computation of the global sum and variance. Additionally, addressing overflow problems requires knowledge of min and max values, as explained in Section 5.2. Afterward, user has the option to apply StandardScaler to the transformed data, requiring global mean and variance.

**Discretization** Binarizer does not require federated computation, as all clients can agree on a preset threshold. On the other hand, KBinsDiscretizer relies on global min and max values to generate intervals with equal width, or it uses quantiles to ensure equal frequency of data samples in each bin. The strategy involving federated $k$-means (see Appendix C) in KBinsDiscretizer needs to update the new clustering centroids during each iteration, which requires compute the global mean of data in each cluster.

**Imputation** In the SimpleImputer, the imputation strategies such as mean, median, and most frequent rely on the aggregation of mean, quantiles, and frequent items sketch. However, more advanced imputers like IterativeImputer (Buck, 1960; Buuren & Groothuis-Oudshoorn, 2011) and KNNImputer (Troyanskaya et al., 2001) require more sophisticated federated algorithms, namely Federated Bayesian Linear Regression (see Appendix D) and Federated $k$-Nearest Neighbors (see Appendix E) for imputing missing values. Notably, the KNN model incorporates a specialized Euclidean distance calculation (Dixon, 1979), which is adapted to handle the presence of missing values in the data.

## 5 NUMERICALLY STABILIZED FEDPOWER

This section first discusses the reasons behind numerical instabilities in power transform. Then, we present our solutions, using the Box-Cox transformation as an example. The federated power transform is outlined in Appendix K.

### 5.1 UNDERSTANDING NUMERICAL INSTABILITIES

Due to the convexity of negative log-likelihood functions, optimizing the parameter $\lambda$ can be achieved through direct minimization or root-finding algorithms. However, these methods involve computing the logarithm of the variance of transformed data, see Equation 3 , which can lead to numerical instabilities when directly squaring large values in the power function.

This numerical instability can affect the optimization, potentially resulting in suboptimal solutions. To illustrate, we apply the exponential search algorithm[2] to two seemingly ordinary datasets. As depicted in Figure 4, the computed result does not match the true minimum.

Additionally, the power transform itself presents a secondary challenge, as depicted in Figure 4. Employing the optimal $\lambda$ for transformation may result in numerical overflow beyond the precision limit. For instance, extreme values such as $2009.0^{104} \approx 3.2 \times 10^{343}$ and $0.1^{-361} = 10^{361}$ can occur. In such cases, users may encounter difficulties analyzing the transformed data or rescaling the Gaussianized data to achieve zero mean and unit variance.

While increasing precision, may partially mitigate these issues, it does not provide a comprehensive solution. Moreover, even seemingly ordinary data, or adversarial data, can exceed double-precision limits, and many mathematical libraries do not support quad-precision or higher due to efficiency considerations.

---

[2]The ExpUpdate algorithm presented in Marchand et al. (2022) contains a typo, which we correct in Appendix F.

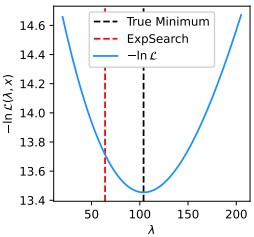
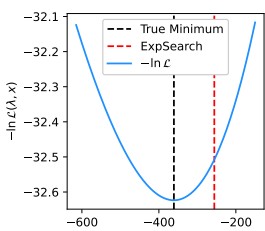

Figure 4: An illustration of the suboptimal results using exponential search on two datasets. The left figure uses data [2003.0, 1950.0, 1997.0, 2000.0, 2009.0], with the true minimum $\lambda \approx 104$; the right figure uses data [0.1, 0.1, 0.1, 0.101], with the true minimum $\lambda \approx -361$. The negative log-likelihood are plotted using the method presented in Section 5.2

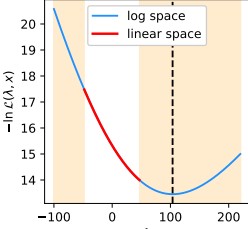
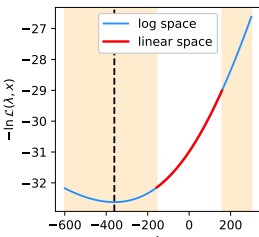

Figure 5: Computation of the negative log-likelihood of Box-Cox as a function of $\lambda$ in log space vs. linear space. The datasets are the same as in Figure 4.

## 5.2 NUMERICALLY STABILIZED POWER TRANSFORM

The primary challenge is to mitigate numerical instabilities and obtain the true minimum during optimization. Notice that directly optimizing the negative log-likelihood function only requires computing the logarithm of the variance on the transformed data. We can leverage computations in the log space to enhance numerical stability, as illustrated in Haberland (2023), which employs the Log-Sum-Exp (LSE) trick (see Appendix G).

A visual representation of this comparison is presented in Figure 5, highlighting the efficacy of log space computations and illustrating the limitations of linear space computations in certain ranges of $\lambda$. Additionally, linear space computations may not be able to find the optimal parameter $\lambda$.

To also better adapt the computation to log space, we carefully chose formulations to ensure numerical stability, particularly when the denominator is near zero, as shown in Figure 6. For Box-Cox transformation, this could also avoid converting some computation into complex domain since $x^\lambda$ is always positive. In particular, when $\lambda \neq 0$, it becomes:

$$\ln \sigma^2_{\psi_{\mathrm{BC}}(\lambda, x)} = \ln \mathrm{Var}[(x^\lambda - 1)/\lambda] \tag{5}$$

$$= \ln \mathrm{Var}(x^\lambda/\lambda) \tag{6}$$

$$= \ln \mathrm{Var}(x^\lambda) - 2\ln|\lambda| \tag{7}$$

To mitigate the transformed data beyond precision limit, we introduce a constraint to confine the transformed data within the representable range of floating-point numbers, specified as $[-y_{\max}, y_{\max}]$. This ensures that positive and negative overflow issues are avoided.

**Lemma 5.1.** *The transformation function $\psi(\lambda, x)$ defined in Equation 1 satisfies the following:*

*(i) $\psi(\lambda, x) \geq 0$ for $x \geq 1$, and $\psi(\lambda, x) < 0$ for $x < 1$.*

*(ii) $\psi(\lambda, x)$ is increasing in both $\lambda$ and $x$.*

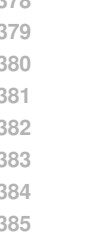

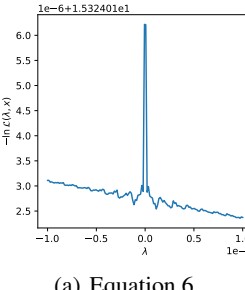
(a) Equation 6

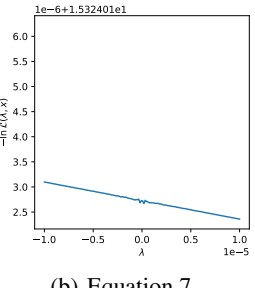
(b) Equation 7

Figure 6: Comparison of methods for calculating the negative log-likelihood of the Box-Cox transformation in log space using Equation 6 and Equation 7, especially when the $\lambda$ approaches zero. The figures use data [2003.0, 1950.0, 1997.0, 2000.0, 2009.0].

Given the Lemma above (proof see Appendix H), we only need to consider at most two points (the minimum and the maximum) to decide the bounds. We formulated the constrained optimization problem below[3].

$$
\begin{aligned}
\min_{\lambda} \quad & -\ln \mathcal{L}_{\mathrm{BC}}(\lambda, x) \\
\text{s.t.} \quad & \text{if } x_{\max} > 1, \lambda \leq \psi_{\mathrm{BC}}^{-1}(x_{\max}, y_{\max}), \\
& \text{if } x_{\min} < 1, \lambda \geq \psi_{\mathrm{BC}}^{-1}(x_{\min}, -y_{\max})
\end{aligned} \tag{8}
$$

Here, $\psi_{\mathrm{BC}}^{-1}$ represents the inverse of Box-Cox to compute $\lambda$ using the Lambert $W$ function (Corless et al., 1996). Given that the solution of $x = a + bc^x$ is $x = a - W(-bc^a \ln c)/\ln c$, the inverse function[4] is defined as:

$$
\psi_{\mathrm{BC}}^{-1}(x, y) = -1/y - W(-x^{-1/y} \ln x/y)/\ln x \tag{9}
$$

Constrained optimization may yield suboptimal results; however, these bounds are crucial to prevent overflow issues and maintain the usability of the transformed data. By default, $y_{\max}$ is set to the maximum value within the floating-point precision of the input data, typically represented as $y_{\max} \approx 10^{308}$ (double-precision). Additionally, users have the flexibility to manually set these bounds, enabling customization based on specific requirements. For instance, setting the bound to infinity can yield optimal unconstrained results, while setting it to a reasonable value prevents the transformed data from becoming excessively large.

## 6 EXPERIMENTAL RESULTS

### 6.1 IMPACT OF DATA PREPROCESSING IN FEDERATED LEARNING

Feature engineering involves various techniques, often rooted in domain-specific knowledge. For tabular data, crucial steps typically include feature scaling, encoding, and handling missing values. In our experiments, we investigate the influence of StandardScaler on the Adult (Becker & Kohavi, 1996), Bank Marketing (Moro et al., 2012), and Covertype (Blackard, 1998) datasets (see Table 5 for the dataset information). Using FedAvg (McMahan et al., 2017) with the SGD optimizer and a Logistic Regression and Multi-Layer Perceptron models, we manually tuned the learning rate from $\{10^{-4}, 3.3 \times 10^{-4}, \ldots, 0.1, 0.33\}$ and report the best result. The data were evenly split in an IID fashion among all clients. The results, illustrated in Table 2 and Appendix L, demonstrate that applying StandardScaler leads to an increase in accuracy ranging from 4% to 37% for Logistic Regression and 2% to 26% for Multi-Layer Perceptron.

---

[3]The constrained optimization for Yeo-Johnson is presented in Appendix I

[4]The Lambert $W$ function, characterized by two branches on the real line, necessitates a subsequent consideration of branch selection (see Appendix J for branch discussion.

Table 2: Test accuracy comparison of FedAvg on Raw vs. Preprocessed Data (StandardScaler) using Logistic Regression (LR) and Multi-Layer Perceptron (MLP).

| Model | # Clients | Preprocessing | Adult | Bank Marketing | Covertype |
|-------|-----------|---------------|-------|----------------|-----------|
| LR | 10 | Raw | 0.792 | 0.777 | 0.529 |
| | | **+Scaling** | **0.824** | **0.893** | **0.725** |
| | 30 | Raw | 0.792 | 0.843 | 0.581 |
| | | **+Scaling** | **0.824** | **0.893** | **0.724** |
| | 100 | Raw | 0.779 | 0.842 | 0.573 |
| | | **+Scaling** | **0.824** | **0.893** | **0.723** |
| | 300 | Raw | 0.775 | 0.829 | 0.550 |
| | | **+Scaling** | **0.824** | **0.892** | **0.723** |
| MLP | 10 | Raw | 0.765 | 0.881 | 0.767 |
| | | **+Scaling** | **0.850** | **0.901** | **0.912** |
| | 30 | Raw | 0.766 | 0.880 | 0.743 |
| | | **+Scaling** | **0.849** | **0.903** | **0.906** |
| | 100 | Raw | 0.764 | 0.880 | 0.696 |
| | | **+Scaling** | **0.850** | **0.902** | **0.877** |
| | 300 | Raw | 0.764 | 0.881 | 0.659 |
| | | **+Scaling** | **0.847** | **0.900** | **0.829** |

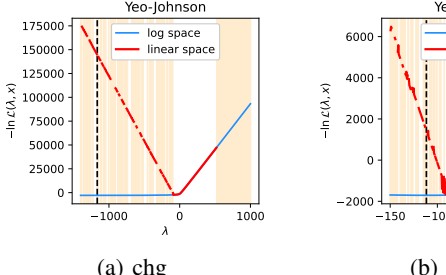 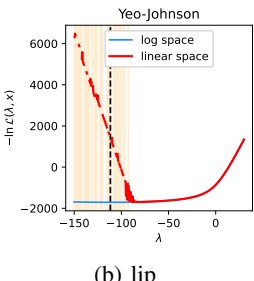

(a) chg  (b) lip

Figure 7: Computation of the negative log-likelihood of Yeo-Johnson as a function of $\lambda$ in log space vs. linear space. The figures use feature chg and lip in the Ecoli dataset.

### 6.2 NUMERICAL EXPERIMENTS ON POWER TRANSFORM

We conduct numerical experiments on three datasets confirmed for their numerical instabilities, as documented in Table 1 of Marchand et al. (2022): Blood Transfusion Service Center (Yeh, 2008), Breast Cancer Wisconsin (Diagnostic) (Wolberg et al., 1995), and Ecoli (Nakai, 1996) (see Table 6 for the dataset information). The objective is to test the computation of the negative log-likelihood function in both log space and linear space. The results are presented in Figure 7 and further detailed in Appendix M.

The computation in linear space may face challenges in identifying the optimal $\lambda$ due to numerical instabilities, see the vertical dotted lines in Figure 7. In contrast, conducting computations in log space not only enables the calculation of the negative log-likelihood over a broader range of $\lambda$ values but also guarantees finding the optimum.

## 7 RELATED WORKS

**Distributed Learning vs. Federated Learning.** Relevant literature includes distributed data preprocessing, where data is centrally stored, and a datacenter performs distributed computation on large-scale datasets. Prior works (Nurmi et al., 2005; Celik et al., 2019) implemented outlier analysis, normalization, and missing value imputation. Spark MLlib (Meng et al., 2016) also offers

Table 3: Federated data preprocessors in *FATE* and *SecretFlow*.

| Frameworks | Preprocessors |
|---|---|
| *FATE ∩ SecretFlow* | MinMaxScaler
StandardScaler
OneHotEncoder
LabelEncoder
KBinsDiscretizer |
| *FATE* | SimpleImputer |
| *SecretFlow* | OrdinalEncoder
LogroundTransformer* |

*A variant of log transformation

a diverse set of functionality for data preprocessing in this setting. Our system is designed for federated learning, where data remains decentralized. Further comparisons between federated and distributed learning can be found in Kairouz et al. (2021).

**Existing Federated Data Preprocessors.** Existing federated learning frameworks, such as *FATE*[5] (Liu et al., 2021) and *SecretFlow*[6] (The SecretFlow Authors, 2022), provide a limited number of preprocessors, summarized in Table 3. Note that we have renamed some preprocessors in *FATE* for better comparison. Additionally, some of their preprocessors have simplified parameters compared to ours, limiting the flexibility of these modules.

**Private Federated Data Preprocessing.** In parallel, there are works on privacy-preserving data preprocessing for federated learning. For example, Hsu and Huang (Hsu & Huang, 2022) implemented one-hot encoding and label encoding based on fully homomorphic encryption. Marchand et al. (Marchand et al., 2022) proposed a private federated Yeo-Johnson based on secure multi-party computation. Given the paramount importance of privacy in federated learning, ensuring that collected statistics do not divulge sensitive information is imperative. And, it's worth noting that quantiles and frequent items inherently contain more information compared to simpler preprocessing techniques like Min/Max scaling. Addressing the privacy implications of these methods remains an area for future research. As outlined in Table 1, the computation can be replaced with their privacy-preserving counterparts, offering enhanced privacy guarantees in federated preprocessing tasks.

## 8 CONCLUSION

In this paper, we highlight the often-underappreciated domain of data preprocessing in Federated Learning, introducing FedPS—a robust suite of tools leveraging aggregated statistics, data sketching, and federated machine learning models. Additionally, we have addressed numerical issues in power transform and proposed a federated version based on Brent's method. By providing a comprehensive and flexible set of data preprocessors, FedPS facilitates the convenient preparation of data, establishing a solid foundation for training federated learning models.

Our future work will delve into privacy-preserving federated data preprocessing, employing techniques like Secure Aggregation (Bonawitz et al., 2017), Secure Multi-Party Computation (Lindell, 2020), and Differential Privacy (Dwork & Roth, 2014). This extension aims to enhance and privacy aspects of FedPS, contributing to the development of more robust federated learning systems.

---

[5]https://fate.readthedocs.io/en/latest/2.0/fate/components/#algorithm-list

[6]https://www.secretflow.org.cn/en/docs/secretflow/v1.9.0b2/source/secretflow.preprocessing

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

## A PREPROCESSORS IN *Scikit-learn*

### A.1 SCALING

Scaling each feature of the data is a common preprocessing step before training machine learning models, referred to as normalization or standardization, usually involving a linear transformation. Different scalers employ various strategies to transform data into predefined ranges. MaxAbsScaler ensures that the maximal absolute value equals 1, MinMaxScaler confines data between a given minimum and maximum value, and StandardScaler ensures that transformed data have a zero mean and unit variance. However, these methods are sensitive to outliers, as the scaling factor depends on them. For more robust scaling, RobustScaler transforms data into a preset quantile range, typically quantiles 0.25 to 0.75 of the data, making it less susceptible to outliers. Notably, Normalizer applies scaling to each data sample instead of each feature, ensuring individual samples have a unit norm, such as $l_1$ (absolute), $l_2$ (euclidean), and max (infinity) norms.

### A.2 ENCODING

Data may contain features represented using strings, necessitating encoding into numeric values. OneHotEncoder encodes them as a one-hot array, and OrdinalEncoder uses an ordinal encoding scheme. They can also ignore infrequent items below a preset threshold of frequency and limit the maximum number of output categories. An alternative approach is the hash trick (Weinberger et al., 2009), exemplified by FeatureHasher, which computes encoding representation based on hash functions. TargetEncoder (Micci-Barreca, 2001) utilizes target mean and the target mean conditioned on the categorical value for encoding, often combined with cross-validation (CV) techniques or additional smoothing parameters to prevent overfitting due to incorporating target information. The default smoothing parameter is set by empirical Bayes estimation, blending the global target variance and the target variance conditioned on the category value. In supervised learning, label encoding might be necessary if labels are strings, addressed by LabelBinarizer for one-vs-all binarization, particularly useful in multiclass classification, and MultiLabelBinarizer in multilabel learning, transforming targets into a multilabel format. LabelEncoder encodes target labels into ordinal numbers, typically used in classification tasks.

### A.3 TRANSFORMATION

Feature transformation is another type of data preprocessing that applies a certain function, typically non-linear, to the features. FunctionTransformer applies a user-defined function to the data, making it useful for tasks like log transformation. PowerTransformer is a parametric method that maps data into a Gaussian distribution, supporting both Box-Cox and Yeo-Johnson transformations. Afterward, the user has the option to apply StandardScaler to the transformed data. QuantileTransformer is a non-parametric method capable of transforming arbitrary data into Gaussian or Uniform distributions. It estimates the cumulative probability distribution function, using quantiles, then maps the data to desired output distributions. SplineTransformer generates univariate B-spline (de Boor, 1978) bases for each feature, particularly useful in time-related feature engineering. It requires setting uniformly distributed knots between the min and max values or along the quantiles.

### A.4 DISCRETIZATION

For continuous features, discretization provides a way to transform them into discrete values, also known as quantization or binning. While it may result in a loss of information, it simplifies the data, making it easier to use and understand. Binarizer uses a threshold to binarize the data. In contrast, KBinsDiscretizer can transform continuous data into $k$ bins using various strategies. It can generate intervals with equal width for each bin or ensure an equal frequency of data samples in each bin. Alternatively, it can employ $k$-means, an unsupervised learning algorithm, to generate $k$ clusters.

### A.5 IMPUTATION

Missing values are prevalent in real-world data for various reasons, posing a challenge for most machine learning algorithms. One common strategy is to discard entire rows or columns containing

missing values, but this approach may introduce bias and reduce the availability of data. Alternatively, imputation strategies, such as SimpleImputer, offer a univariate method to fill missing values with the mean, median, mode (most frequent item) of the respective feature, or a constant value. In contrast, IterativeImputer (Buck, 1960; Buuren & Groothuis-Oudshoorn, 2011), a multivariate imputation strategy, models missing values as a function of other features. It selects a specific feature column as the target and utilizes other features as inputs to fit a regression model, subsequently using this model to predict the missing values. Another method, KNNImputer (Troyanskaya et al., 2001), employs a weighted average of $k$-nearest neighbors for imputation.

## B    SUPPLEMENTAL TABLES

Below is the aggregated statistics and corresponding preprocessors, along with an analysis of the communication cost from the client's perspective. Assume each client has a dataset $\mathbf{X}$ containing $n$ examples (rows) and $m$ features (columns), and the dataset $\mathbf{X}'$ on which preprocessing steps will be applied contains $n'$ examples and $m'$ features. For iterative processes or algorithms such as k-Means, PowerTransformer, and IterativeImputer, we denote the number of iterations as $t$. In the case of k-Means, k Nearest Neighbors, and frequent items sketch (with $k$ bins), $k$ represents different values depending on the specific context in the communication cost analysis. For encoding methods, we assume there are $d$ distinct categories across $n$ examples. Lastly, the KLL sketch (Karnin et al., 2016) is employed as the default quantile sketch, and its communication cost (space usage) is referenced from that work.

## C    FEDERATED K-MEANS

The $k$-Means clustering is a unsupervised algorithm, which find $k$ cluster centroids $\{\mu_1, \ldots, \mu_k\}$. It is an iterative algorithm that first computes distances between each data sample and each cluster centroid. Afterwards, it assign each data sample $x_i$ to its closest cluster $S_j$. Finally, each cluster centroid is updated by the mean of data samples in each cluster, i.e., $\mu_j = \sum_{x_i \in S_j} x_i / n_j$. For horizontal federated $k$-Means, see Algorithm 1, the server needs to first broadcast the cluster centroids to all clients, then compute the global mean of data samples in each cluster to update new cluster centroids. The colored steps indicate communication between the server and clients, with blue indicating receiving and orange indicating sending actions.

---

**Algorithm 1:** Horizontal Federated k-Means (Server)

---

**Input:** Client $c$ has data $\{x_i^{(c)}\}$

1  Initialize clustering centroids $\{\mu_1, \ldots, \mu_k\}$

2  **repeat**

3      Broadcast clustering centroids $\{\mu_1, \ldots, \mu_k\}$ to all clients

    `// Each client assign each data sample` $x_i^{(c)}$ `to its closest`
        `cluster` $S_j$

4      Collect the local sums in each cluster $\{s_1^{(c)}, s_2^{(c)}, \ldots, s_k^{(c)}\}$ where $s_j^{(c)} = \sum_{x_i^{(c)} \in S_j} x_i^{(c)}$

    and the number of data samples in each cluster $\{n_1^{(c)}, \ldots, n_k^{(c)}\}$

    Set new clustering centroids $\mu_j = \sum_c s_j^{(c)} / \sum_c n_j^{(c)}$

5  **until** *Convergence or reach the max iteration*;

**Output:** Clustering centroids $\{\mu_1, \ldots, \mu_k\}$

---

## D    FEDERATED BAYESIAN LINEAR REGRESSION

Bayesian Linear Regression adopts a probabilistic approach to define the model parameters. Typically, the model parameters are assumed to follow a zero-mean isotropic Gaussian distribution Tipping (2001); Bishop (2006) as given by:

$$p(\boldsymbol{\omega}) = \mathcal{N}(\boldsymbol{\omega}|\mathbf{0}, \alpha^{-1}\mathbf{I}) \tag{10}$$

Table 4: Aggregated statistics and associated preprocessors.

| Aggregated Statistics | Associated Preprocessors | Communication Cost (Client) |
|---|---|---|
| Min, Max | MaxAbsScaler* | $O(m)$ |
| | MinMaxScaler | $O(m)$ |
| | Normalizer (max norm)* | $O(n)$ |
| | KBinsDiscretizer (strategy=uniform) | $O(m)$ |
| | PowerTransformer | $O(m)$ |
| | SplineTransformer (knots=uniform) | $O(m)$ |
| | KNNImputer (Horizontal)† | $O(n'km)$ |
| Sum | Normalizer ($l_1$ or $l_2$ norm) | $O(n)$ |
| | PowerTransformer | $O(m)$ |
| | KNNImputer (Vertical) | $O(n'n)$ |
| | IterativeImputer (Horizontal) | $O(tm^2 \min(n, m))$ |
| | IterativeImputer (Vertical) | $O(tmn \min(n, m))$ |
| Mean | StandardScaler (with_mean=True) | $O(m)$ |
| | SimpleImputer (strategy=mean) | $O(m)$ |
| | TargetEncoder | $O(dm)$ |
| | PowerTransformer (standardize=True) | $O(m)$ |
| | KBinsDiscretizer (strategy=kmeans) | $O(tkm)$ |
| | KNNImputer (Horizontal) | $O(n'km)$ |
| Variance | StandardScaler (with_std=True) | $O(m)$ |
| | TargetEncoder | $O(dm)$ |
| | PowerTransformer | $O(tm)$ |
| Quantiles | RobustScaler | |
| | KBinsDiscretizer (strategy=quantile) | |
| | QuantileTransformer | $O(\frac{1}{\epsilon} \log^2 \log \frac{1}{\epsilon} \cdot m)$ |
| | SplineTransformer (knots=quantile) | |
| | SimpleImputer (strategy=median) | |
| Set Union | LabelBinarizer | |
| | MultiLabelBinarizer | $O(d)$ |
| | LabelEncoder | |
| | OneHotEncoder | |
| | OrdinalEncoder | $O(dm)$ |
| | TargetEncoder | |
| Frequent items | OneHotEncoder (group infrequent categories) | |
| | OrdinalEncoder (group infrequent categories) | $O(km)$ |
| | SimpleImputer (strategy=most_frequent) | |

*Max only, †Min only

The posterior distribution of the parameters takes the form of a Gaussian distribution:

$$p(\boldsymbol{\omega}|\mathbf{X}, \mathbf{Y}, \beta) = \mathcal{N}(\boldsymbol{\omega}|\hat{\boldsymbol{\omega}}, \boldsymbol{\Sigma}) \tag{11}$$

where:

$$\hat{\boldsymbol{\omega}} = \beta \boldsymbol{\Sigma}^{-1} \mathbf{X}^T \mathbf{Y} \tag{12}$$

$$\boldsymbol{\Sigma} = \alpha \mathbf{I} + \beta \mathbf{X}^T \mathbf{X} \tag{13}$$

The hyperparameters $\alpha$ and $\beta$ can be modeled using Gamma distributions as hyperpriors:

$$p(\alpha) = \text{Gamma}(\alpha|a_1, a_2) \tag{14}$$

$$p(\beta) = \text{Gamma}(\beta|b_1, b_2) \tag{15}$$

An iterative process is used to re-estimate the hyperparameters $\alpha$ and $\beta$, followed by updates to $\hat{\omega}$ and $\Sigma$:

$$\alpha = \frac{n - \gamma + 2a_1}{\varepsilon + 2a_2} \tag{16}$$

$$\beta = \frac{\gamma + 2b_1}{\|\hat{\omega}\|_2^2 + 2b_2} \tag{17}$$

$$\gamma = \sum_i \frac{\alpha \Lambda_i}{\beta + \alpha \Lambda_i} \tag{18}$$

$$\varepsilon = \|\mathbf{Y} - \mathbf{X}\hat{\omega}\|_2^2 \tag{19}$$

To compute the matrix inverse in Equation 13 efficiently, Singular Value Decomposition (SVD) is applied:

$$\Sigma^{-1} = \mathbf{V}(\frac{1}{\alpha}\mathbf{I} + \frac{1}{\beta}\Lambda^{-1})\mathbf{V}^T \tag{20}$$

where $\mathbf{U}, \mathbf{S}, \mathbf{V}^T = \text{SVD}(\mathbf{X})$ and $\mathbf{S}^2 = \Lambda$.

In Horizontal Bayesian Linear Regression, since the data is partitioned by examples, the server can aggregate the terms $\mathbf{X}^T\mathbf{X}$ from each client:

$$\mathbf{X}^T\mathbf{X} = \begin{bmatrix} \mathbf{X}^{(1)^T} & \mathbf{X}^{(2)^T} & \cdots \end{bmatrix} \begin{bmatrix} \mathbf{X}^{(1)} \\ \mathbf{X}^{(2)} \\ \vdots \end{bmatrix} = \mathbf{X}^{(1)^T}\mathbf{X}^{(1)} + \mathbf{X}^{(2)^T}\mathbf{X}^{(2)} + \ldots \tag{21}$$

---

**Algorithm 2:** Horizontal Federated Bayesian Linear Regression (Server)

**Input:** Client $c$ has data $\mathbf{X}^{(c)}$ and $\mathbf{Y}^{(c)}$

1 Initialize $\alpha$ and $\beta$

2 Compute global sum $\mathbf{X}^T\mathbf{Y} = \sum_c \mathbf{X}^{(c)^T}\mathbf{Y}^{(c)}$

3 Compute global sum $\mathbf{X}^T\mathbf{X} = \sum_c \mathbf{X}^{(c)^T}\mathbf{X}^{(c)}$

4 Compute eigenvalues $\Lambda$ and eigenvectors $\mathbf{V}$ of $\mathbf{X}^T\mathbf{X}$

5 **repeat**

6     Compute $\Sigma^{-1} = \mathbf{V}(\frac{1}{\alpha}\mathbf{I} + \frac{1}{\beta}\Lambda^{-1})\mathbf{V}^T$

7     Compute $\hat{\omega} = \beta\Sigma^{-1}\mathbf{X}^T\mathbf{Y}$

8     Broadcast the model parameter $\hat{\omega}$ to all clients

9     Compute the global error $\varepsilon$

10     Update $\alpha$ and $\beta$

11 **until** *Convergence or reach the max iteration*;

**Output:** model parameter $\hat{\omega}$

---

For Vertical Bayesian Linear Regression, where data is split by features, the formulas are adjusted as follows:

$$\hat{\omega} = \beta\mathbf{X}^T\Sigma^{-1}\mathbf{Y} \tag{22}$$

$$\Sigma = \alpha\mathbf{I} + \beta\mathbf{X}\mathbf{X}^T \tag{23}$$

Here, the server sums over the feature matrices:

$$\mathbf{X}\mathbf{X}^T = \begin{bmatrix} \mathbf{X}^{(1)} & \mathbf{X}^{(2)} & \cdots \end{bmatrix} \begin{bmatrix} \mathbf{X}^{(1)^T} \\ \mathbf{X}^{(2)^T} \\ \vdots \end{bmatrix} = \mathbf{X}^{(1)}\mathbf{X}^{(1)^T} + \mathbf{X}^{(2)}\mathbf{X}^{(2)^T} + \ldots \tag{24}$$

To compute the matrix inverse in Equation 23, SVD can again be employed:

$$\Sigma^{-1} = \mathbf{U}(\frac{1}{\alpha}\mathbf{I} + \frac{1}{\beta}\Lambda^{-1})\mathbf{U}^T \tag{25}$$

---

**Algorithm 3:** Vertical Federated Bayesian Linear Regression (Server)

---

**Input:** Client $c$ has data $\mathbf{X}^{(c)}$ and only one client has $\mathbf{Y}$

1  Initialize $\alpha$ and $\beta$

2  Receive $\mathbf{Y}$ from the client who has target

3  Compute global sum $\mathbf{X}\mathbf{X}^T = \sum_c \mathbf{X}^{(c)}\mathbf{X}^{(c)T}$

4  Compute eigenvalues $\mathbf{\Lambda}$ and eigenvectors $\mathbf{U}$ of $\mathbf{X}\mathbf{X}^T$

5  **repeat**

6      Compute $\mathbf{\Sigma}^{-1} = \mathbf{U}(\frac{1}{\alpha}\mathbf{I} + \frac{1}{\beta}\mathbf{\Lambda}^{-1})\mathbf{U}^T$

7      Broadcast $\beta\mathbf{\Sigma}^{-1}\mathbf{Y}$ to all clients

     // Each client compute $\hat{\boldsymbol{\omega}}^{(c)} = \mathbf{X}^{(c)T}\beta\mathbf{\Sigma}^{-1}\mathbf{Y}$

8      Compute the global prediction $\hat{\mathbf{Y}} = \sum_c \mathbf{X}^{(c)}\hat{\boldsymbol{\omega}}^{(c)}$

9      Compute the error $\varepsilon = \|\mathbf{Y} - \hat{\mathbf{Y}}\|_2^2$

10      Compute the global sum $\|\hat{\boldsymbol{\omega}}\|_2^2 = \sum_c \|\hat{\boldsymbol{\omega}}^{(c)}\|_2^2$

11      Update $\alpha$ and $\beta$

12  **until** *Convergence or reach the max iteration*;

    **Output:** model parameter $\hat{\boldsymbol{\omega}}$

---

# E  FEDERATED k NEAREST NEIGHBORS REGRESSION

The $k$ Nearest Neighbors ($k$NN) regression is a non-parametric algorithm that identifies the $k$ closest examples to a given point $x$ and then averages the target values $y$ of these neighbors. A commonly used distance metric is the Euclidean distance. The averaging can be done using either the ordinary mean or a weighted mean, where the weights are the reciprocals of the distances.

For horizontal federated $k$NN regression (Khedr, 2008), each client computes its local top-$k$ minimum distances and sends these distances to the server, as illustrated in Algorithm 4. The server then determines the global $k$ nearest neighbors and retrieves their corresponding target values to compute the average.

---

**Algorithm 4:** Horizontal Federated kNN Regression (Server)

---

**Input:** Client $c$ has data $\{x_i^{(c)}, y_i^{(c)}\}$, data $x_p$ need to be predicted

1  Broadcast data $x_p$ to all clients

2  Collect local top-$k$ minimum distances $\{d_1^{(c)}, \ldots, d_k^{(c)}\}$ between $x_p$ and each client's data

3  Compute the global top-$k$ minimum distances $\{d_1, \ldots, d_k\}$ and their indices

4  Send the indices of $k$ nearest neighbors to their corresponding clients

5  Compute (weighted) mean $\mu$ of the target $y$ based on the indices

    **Output:** (Weighted) mean $\mu$

---

For vertical federated $k$NN regression, described in Algorithm 5, the distance cannot be directly computed since clients possess different features. In this case, each client computes one segment of the distance and sends it to the server. The server sums over each distance segment to identify the global $k$ nearest neighbors. Finally, the server sends the indices of these neighbors to the client requesting the prediction.

---

**Algorithm 5:** Vertical Federated kNN Regression (Server)

---

**Input:** Client $c$ has data $\{x_i^{(c)}, y_i^{(c)}\}$, data $x_p^{(c)}$ need to be predicted

    // Each client compute the local distance between $x_p^{(c)}$ and each
        data sample

**1** Compute the global distance between $x_p$ and each data sample

**2** Select the global top-k minimum distances $\{d_1, \ldots, d_k\}$ and their indices

**3** Send the indices of $k$ nearest neighbors to client whose data contain target

    // The client compute (weighted) mean $\mu$ of the target $y$ based
        on the indices

**Output:** (Weighted) mean $\mu$

---

## F  CORRECTION OF THE EXPUPDATE ALGORITHM

The original ExpUpdate algorithm proposed in (Marchand et al., 2022) contains a typo that causes the algorithm to update in the opposite direction. We present the corrected version below, with the modified part highlighted in red.

---

**Algorithm 6:** ExpUpdate

---

**Input:** $\lambda, \lambda^+, \lambda^-, \Delta \in \{-1, 1\}$

**1** **if** $\Delta = -1$ **then**

**2**     $\lambda^- \leftarrow \lambda$

**3**     **if** $\lambda^+ < \infty$ **then**

**4**         $\lambda \leftarrow (\lambda^+ + \lambda)/2$

**5**     **else**

**6**         $\lambda \leftarrow \max(2\lambda, 1)$

**7**     **end**

**8** **else**

**9**     $\lambda^+ \leftarrow \lambda$

**10**     **if** $\lambda^- > -\infty$ **then**

**11**         $\lambda \leftarrow (\lambda^- + \lambda)/2$

**12**     **else**

**13**         $\lambda \leftarrow \min(2\lambda, -1)$

**14**     **end**

**15** **end**

**Output:** Updated $\lambda, \lambda^+, \lambda^-$

---

## G  LOG SPACE COMPUTATION VIA THE LSE TRICK

The LSE trick is defined as follows:

$$
\begin{aligned}
\mathrm{LSE}(x_1, \ldots, x_n) &= \ln \sum \exp(x_i) \\
&= \ln \sum \exp(x_i - c) + c \\
&\text{where } c = \max x_i
\end{aligned}
\tag{26}
$$

As illustrated by (Haberland, 2023), this trick enables the computation of the logarithmic mean and variance, mitigating potential numerical overflow issues. For instance, Equation 27 employs LSE to compute the logarithmic mean term.

$$
\begin{aligned}
\ln \mu &= \ln \sum x_i / n \\
&= \ln \sum x_i - \ln n \\
&= \mathrm{LSE}(\ln x_1, \ldots, \ln x_n) - \ln n
\end{aligned}
\tag{27}
$$

The logarithmic variance term, as outlined in Equation 29, involves the LSE first applied to the logarithms of the squared differences, preventing numerical overflow in comparison to standard linear space computations.

$$\ln(x_i - \mu) = \ln[\exp(\ln x_i) + \exp(\ln \mu + \pi i)]$$
$$= \text{LSE}(\ln x_i, \ln \mu + \pi i) \tag{28}$$
$$\text{where } \pi i \text{ is the imaginary part.}$$

$$\ln \sigma^2 = \ln \sum (x_i - \mu)^2 - \ln n$$
$$= \text{LSE}[2\ln(x_1 - \mu), \ldots, 2\ln(x_n - \mu)] - \ln n \tag{29}$$

## H  PROPERTIES OF THE BOX-COX TRANSFORMATION FUNCTION

The Box-Cox transformation function has properties similar to Yeo-Johnson, as described in (Yeo, 1997; 2000).

**Lemma H.1.** *The transformation function $\psi(\lambda, x)$ defined in Equation 1 satisfies the following:*

*(i) $\psi(\lambda, x) \geq 0$ for $x \geq 1$, and $\psi(\lambda, x) < 0$ for $x < 1$.*

*(ii) $\psi(\lambda, x)$ is convex in $x$ for $\lambda > 1$ and concave in $x$ for $\lambda < 1$.*

*(iii) $\psi(\lambda, x)$ is a continuous function of $(\lambda, x)$.*

*(iv) If $\psi^{(k)} = \partial^k \psi(\lambda, x)/\partial \lambda^k$ then, for $k \geq 1$.*

$$\psi^{(k)} = \begin{cases} [x^\lambda (\ln x)^k - k\psi^{(k-1)}]/\lambda & \text{if } \lambda \neq 0, \\ (\ln x)^{k+1}/(k+1) & \text{if } \lambda = 0. \end{cases}$$

$\psi^{(k)}$ *is continuous in $(\lambda, x)$ and $\psi^{(0)} \equiv \psi(\lambda, x)$.*

*(v) $\psi(\lambda, x)$ is increasing in both $\lambda$ and $x$.*

*(vi) $\psi(\lambda, x)$ is convex in $\lambda$ for $x > 1$ and concave in $\lambda$ for $0 < x < 1$.*

*Proof.*    (i) For $x \geq 1$, we have
$$\begin{cases} x^\lambda - 1 \geq 0 & \text{if } \lambda > 0, \\ x^\lambda - 1 \leq 0 & \text{if } \lambda < 0. \end{cases}$$

When $\lambda = 0$, $\ln(x) \geq 0$ for $x \geq 1$. Hence $\psi(\lambda, x) \geq 0$ for all $\lambda$ whenever $x \geq 1$. Similarly, for $0 < x < 1$, we have
$$\begin{cases} x^\lambda - 1 < 0 & \text{if } \lambda > 0, \\ x^\lambda - 1 > 0 & \text{if } \lambda < 0. \end{cases}$$

When $\lambda = 0$, $\ln(x) < 0$ for $0 < x < 1$. Hence $\psi(\lambda, x) < 0$ for all $\lambda$ whenever $0 < x < 1$.

(ii) The second order partial derivative of $\psi$ respect to $x$ is

$$\frac{\partial^2 \psi(\lambda, x)}{\partial x^2} = \begin{cases} (\lambda - 1)x^{\lambda-2} & \text{if } \lambda \neq 0, \\ -1/x^2 & \text{if } \lambda = 0. \end{cases}$$

Therefore, $\frac{\partial^2 \psi(\lambda,x)}{\partial x^2} > 0$ when $\lambda > 1$ and $\frac{\partial^2 \psi(\lambda,x)}{\partial x^2} < 0$ when $\lambda < 1$.

(iii) It's clear that $\psi(\lambda, x)$ is continuous for $\lambda$ and $x$ except $\lambda = 0$. We just need to prove it's continuous at $\lambda = 0$. By L'Hopital's rule, we have

$$\lim_{\lambda \to 0} \frac{x^\lambda - 1}{\lambda} = \lim_{\lambda \to 0} \frac{x^\lambda \ln x}{1} = \ln x$$

(iv) We prove this by induction. Let $k = 1$, then for $\lambda \neq 0$

$$\psi^{(1)} = \frac{x^\lambda \lambda \ln x - (x^\lambda - 1)}{\lambda^2} = \frac{x^\lambda \ln x - \psi^{(0)}}{\lambda}$$

For $\lambda = 0$, by L'Hopital's rule, we have

$$\begin{aligned}
\psi^{(1)}(0, x) &= \lim_{\lambda \to 0} \frac{\psi(\lambda, x) - \psi(0, x)}{\lambda} \\
&= \lim_{\lambda \to 0} \psi^{(1)}(\lambda, x) \\
&= \lim_{\lambda \to 0} \frac{x^\lambda \lambda \ln x - x^\lambda + 1}{\lambda^2} \\
&= \lim_{\lambda \to 0} \frac{x^\lambda (\ln x)^2}{2} \\
&= (\ln x)^2 / 2
\end{aligned}$$

Assume that this hold for $k = n$ where $n \geq 1$, then for $k = n + 1$ and $\lambda \neq 0$

$$\begin{aligned}
\psi^{(n+1)} &= \frac{\partial}{\partial \lambda} \frac{x^\lambda (\ln x)^n - n\psi^{(n-1)}}{\lambda} \\
&= \frac{[x^\lambda (\ln x)^{n+1} - n\psi^{(n)}]\lambda - [x^\lambda (\ln x)^n - n\psi^{(n-1)}]}{\lambda^2} \\
&= \frac{x^\lambda (\ln x)^{n+1} - (n+1)\psi^{(n)}}{\lambda}
\end{aligned}$$

For $\lambda = 0$, by L'Hopital's rule, we have

$$\begin{aligned}
\psi^{(n+1)}(0, x) &= \lim_{\lambda \to 0} \frac{\psi^{(n)}(\lambda, x) - \psi^{(n)}(0, x)}{\lambda} \\
&= \lim_{\lambda \to 0} \psi^{(n+1)}(\lambda, x) \\
&= \lim_{\lambda \to 0} \frac{x^\lambda (\ln x)^{n+1} - (n+1)\psi^{(n)}}{\lambda} \\
&= \lim_{\lambda \to 0} x^\lambda (\ln x)^{n+2} - (n+1)\psi^{(n+1)}(\lambda, x) \\
&= (\ln x)^{n+2} - (n+1) \lim_{\lambda \to 0} \psi^{(n+1)}(\lambda, x)
\end{aligned}$$

Therefore, $\psi^{(n+1)}(0, x) = \lim_{\lambda \to 0} \psi^{(n+1)}(\lambda, x) = (\ln x)^{n+2} / (n+2)$

Thus, the recurrence relation holds for all $k \geq 1$ and $\lambda \neq 0$.

(v) The partial derivative of $\psi$ respect to $x$ is

$$\frac{\partial \psi(\lambda, x)}{\partial x} = \begin{cases} x^{\lambda - 1} & \text{if } \lambda \neq 0, \\ 1/x & \text{if } \lambda = 0. \end{cases}$$

so $\frac{\partial \psi(\lambda, x)}{\partial x} > 0$. Therefore, $\psi$ is increasing in $x$.

The partial derivative of $\psi$ respect to $\lambda$ is

$$\frac{\partial \psi(\lambda, x)}{\partial \lambda} = \begin{cases} \frac{x^\lambda (\ln x^\lambda - 1) + 1}{\lambda^2} & \text{if } \lambda \neq 0, \\ (\ln x)^2 / 2 & \text{if } \lambda = 0. \end{cases}$$

Let $y = x^\lambda > 0$ and $f_1(y) = y(\ln y - 1) + 1$, we have $f_1'(y) = \ln y$, $f_1''(y) = 1/y > 0$. Thus $f_1(y)$ has the unique minimum at $y = 1$ and $f_1(y) > f_1(1) = 0$. Thus $\frac{\partial \psi(\lambda, x)}{\partial \lambda} > 0$. Therefore, $\psi$ is increasing in $\lambda$.

(vi) The second order partial derivative of $\psi$ respect to $\lambda$ is

$$\frac{\partial^2 \psi(\lambda, x)}{\partial \lambda^2} = \begin{cases} \frac{x^\lambda[(\ln x^\lambda)^2 - 2\ln x^\lambda + 2] - 2}{\lambda^3} & \text{if } \lambda \neq 0, \\ (\ln x)^3/3 & \text{if } \lambda = 0. \end{cases}$$

Let $y = x^\lambda > 0$ and $f_2(y) = y[(\ln y)^2 - 2\ln y + 2] - 2$, we have $f_2'(y) = (\ln y)^2 > 0$ and $f_2(1) = 0$. Thus $f_2(y) > 0$ when $y > 1$ and $f_2(y) < 0$ when $y < 1$ since $f_2(y)$ is increasing in $y$.

The relationship between $x, \lambda$ and $y, f_2(y)$ are as follows

$$\begin{rcases} x > 1, \lambda > 0 \quad \Rightarrow y > 1, f_2(y) > 0 \\ x > 1, \lambda < 0 \quad \Rightarrow y < 1, f_2(y) < 0 \end{rcases} \Rightarrow f_2(y)/\lambda^3 > 0$$

$$\begin{rcases} 0 < x < 1, \lambda < 0 \quad \Rightarrow y > 1, f_2(y) > 0 \\ 0 < x < 1, \lambda > 0 \quad \Rightarrow y < 1, f_2(y) < 0 \end{rcases} \Rightarrow f_2(y)/\lambda^3 < 0$$

Therefore, $\frac{\partial^2 \psi(\lambda, x)}{\partial \lambda^2} > 0$ when $x > 1$ and $\frac{\partial^2 \psi(\lambda, x)}{\partial \lambda^2} < 0$ when $0 < x < 1$.

$\square$

## I  THE CONSTRAINED OPTIMIZATION FOR YEO-JOHNSON

Utilizing the properties of the Yeo-Johnson transformation function (refer to Lemma 1 in (Yeo, 2000)), the constrained optimization is similar to Equation 8, with a distinction at the point of sign change at $x = 0$.

$$\begin{aligned} \min_\lambda \quad & -\ln \mathcal{L}_{\text{YJ}}(\lambda, x) \\ \text{s.t.} \quad & \text{if } x_{\max} > 0, \lambda \leq \psi_{\text{YJ}}^{-1}(x_{\max}, y_{\max}), \\ & \text{if } x_{\min} < 0, \lambda \geq \psi_{\text{YJ}}^{-1}(x_{\min}, -y_{\max}) \end{aligned} \tag{30}$$

Using the Lambert $W$ function, the inverse function to compute $\lambda$ is defined as follows:

$$\psi_{\text{YJ}}^{-1}(x, y) = \begin{cases} -1/y - W\left(-\frac{(x+1)^{-1/y}\ln(x+1)}{y}\right)/\ln(x+1) & \text{if } x \geq 0, \\ 2 - 1/y + W\left(\frac{(1-x)^{1/y}\ln(1-x)}{y}\right)/\ln(1-x) & \text{if } x < 0. \end{cases} \tag{31}$$

## J  THE CHOICE OF TWO REAL BRANCHES IN THE LAMBERT $W$ FUNCTION

The constrained optimization relies on the inverse functions of Box-Cox (see Equation 9) and Yeo-Johnson (see Equation 31) to determine the constrained value of $\lambda$. However, the Lambert $W$ function has two real branches: the $k = 0$ branch for $W(x) \geq -1$ and the $k = -1$ branch for $W(x) \leq -1$ (Corless et al., 1996).

Here, we use the inverse function of Box-Cox to illustrate the choice of $k$; the Yeo-Johnson analysis is analogous and thus omitted. When overflows occur during the transformation, and both $y$ and $x^\lambda$ approach the largest representable floating-point number, we can express Equation 9 differently:

$$W(-x^{-1/y}\ln x/y) = -(\lambda + 1/y)\ln x \approx -\lambda \ln x = -\ln x^\lambda \ll -1 \tag{32}$$

As a result, the $k = -1$ branch should be used for computing the upper and lower bounds for $\lambda$.

## K  FEDERATED POWER TRANSFORM

The FedPower, outlined in Algorithm 7, a federated algorithm designed for power transformations. The algorithm comprises two primary steps: (1) Address numerical issues, as detailed in Section 5.2. (2) Conduct minimization using Brent's method.

1. Compute the constraint for $\lambda$.

   As depicted in Equation 8, constrained optimization ensures that the transformed data falls within the representable range of floating-point numbers. Consequently, it is essential to compute both the global minimum and maximum to establish upper and lower bounds for $\lambda$.

2. Perform minimization via Brent's method.

   This step involves evaluating the negative log-likelihood function at various $\lambda$ points. The log-likelihood function can be divided into two parts. The summation part depends solely on the data $x$, requiring a one-time computation that can be cached. However, the log-variance part is $\lambda$-dependent, necessitating an iterative approach for aggregation. This computation is also performed in the log space.

---

**Algorithm 7:** FedPower (Server)

**Input:** Data $\{x_i\}$ distributed at clients

1  Compute total data size $n$
2  Compute the global $x_{\min}$ and $x_{\max}$
3  Compute the constraint $[\lambda_L, \lambda_U]$ for $\lambda$
4  Compute the global sum: $\sum_i^n \ln x_i$ (BC) or $\sum_i^n \text{sgn}(x_i) \ln(|x_i| + 1)$ (YJ)
   `// Start Brent's method`
5  **repeat**
6  $\quad$ Broadcast the candidate $\lambda_c$ to all clients
7  $\quad$ Compute the global log-variance $\ln \sigma^2_{\psi(\lambda_c, x)}$
8  $\quad$ Compute the negative log-likelihood $-\ln \mathcal{L}(\lambda_c, x)$
9  $\quad$ Continue Brent's method
10 **until** *Convergence or reach the max iteration*;
   **Output:** The constrained optimal $\lambda^*$

---

Compared to the exponential search used in Marchand et al. (2022), which exhibits a linear convergence rate, our proposed method achieves superlinear convergence, a key benefit inherent to Brent's methods.

## L    SUPPLEMENTAL TABLES AND FIGURES FOR THE EFFECT OF DATA PREPROCESSING

Table 5: Dataset information for the data preprocessing experiment.

| Datasets | # Train Instances | # Test Instances | # Features | # Classes |
|---|---|---|---|---|
| Adult | 32561 | 16281 | 14 | 2 |
| Bank Marketing | 31647 | 13564 | 16 | 2 |
| Covertype | 406708 | 174304 | 54 | 7 |

## M    SUPPLEMENTAL TABLES AND FIGURES FOR NUMERICAL EXPERIMENTS ON POWER TRANSFORM

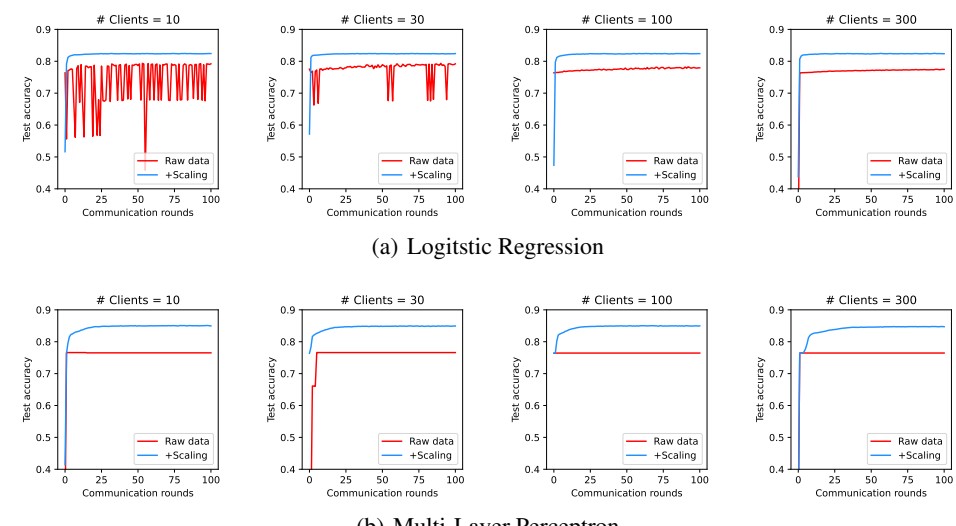

(a) Logitstic Regression

(b) Multi-Layer Perceptron

Figure 8: Test accuracy comparison of FedAvg on Raw vs. Preprocessed data using Logistic Regression and Multi-Layer Perceptron on the Adult dataset.

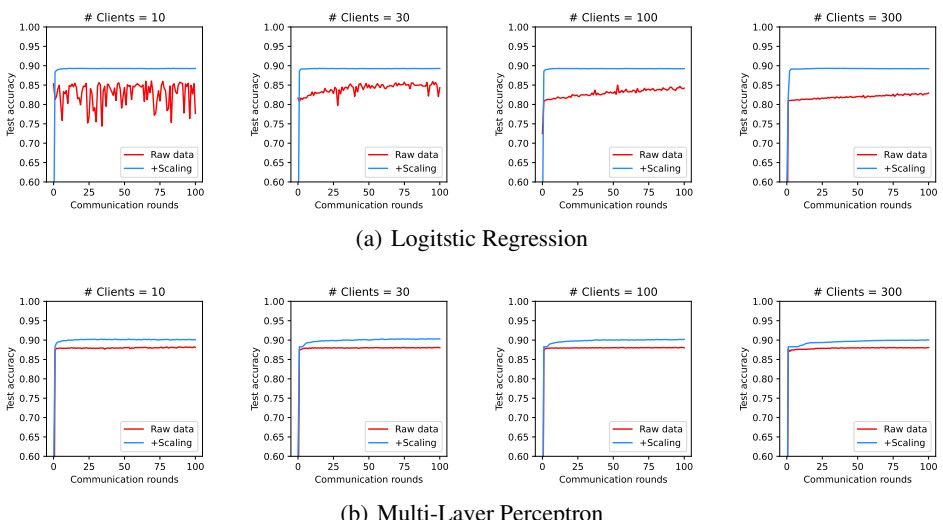

(a) Logitstic Regression

(b) Multi-Layer Perceptron

Figure 9: Test accuracy comparison of FedAvg on Raw vs. Preprocessed data using Logistic Regression and Multi-Layer Perceptron on the Bank Marketing dataset.

Table 6: Dataset information for the numerical experiment.

| Datasets | # Instances | # Numeric Features | # Strictly Positive Numeric Features |
|---|---|---|---|
| Blood Transfusion Service Center | 748 | 5 | 3 |
| Breast Cancer Wisconsin (Diagnostic) | 569 | 31 | 25 |
| Ecoli | 336 | 7 | 4 |

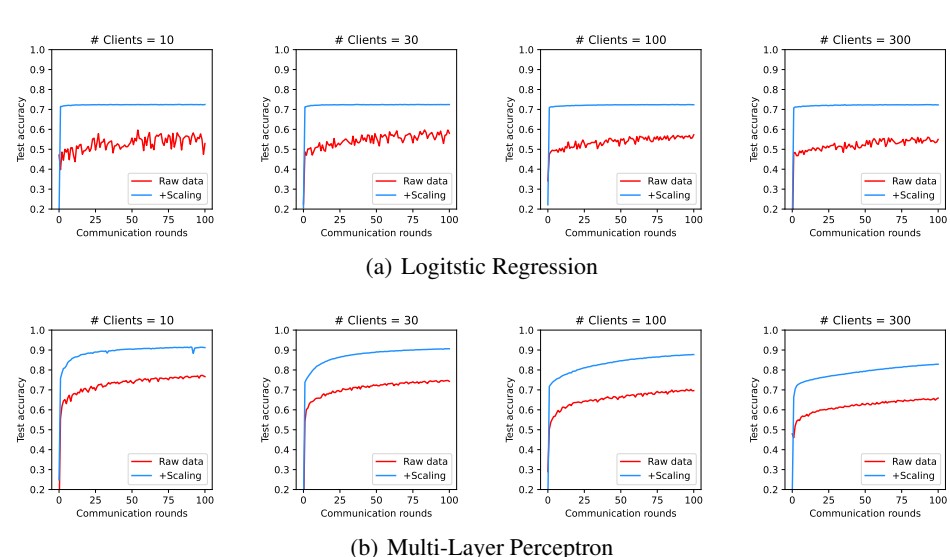

Figure 10: Test accuracy comparison of FedAvg on Raw vs. Preprocessed data using Logistic Regression and Multi-Layer Perceptron on the Covertype dataset.

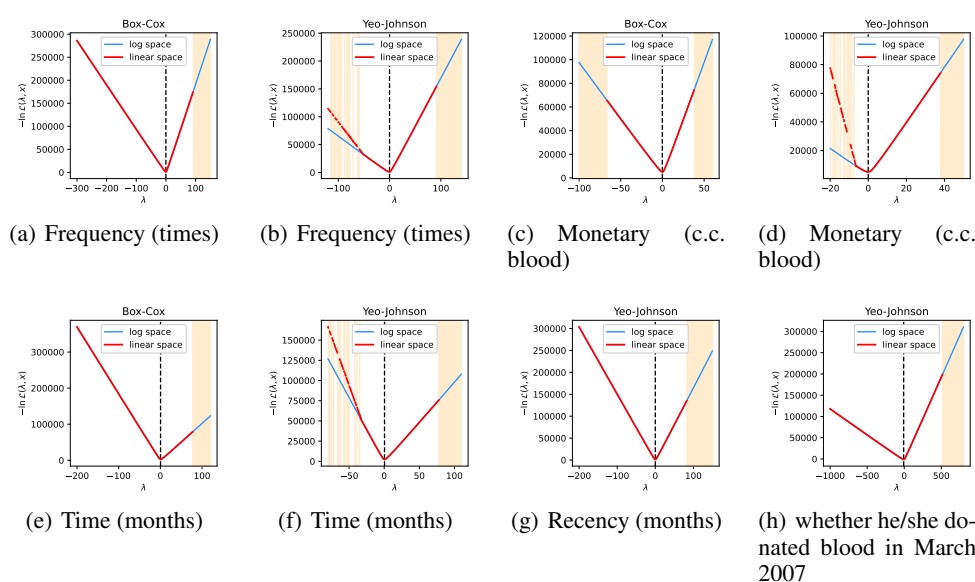

Figure 11: Computation of the negative log-likelihood of Box-Cox and Yeo-Johnson as a function of $\lambda$ in log space vs. linear space. The figures use features in the Blood Transfusion Service Center dataset.

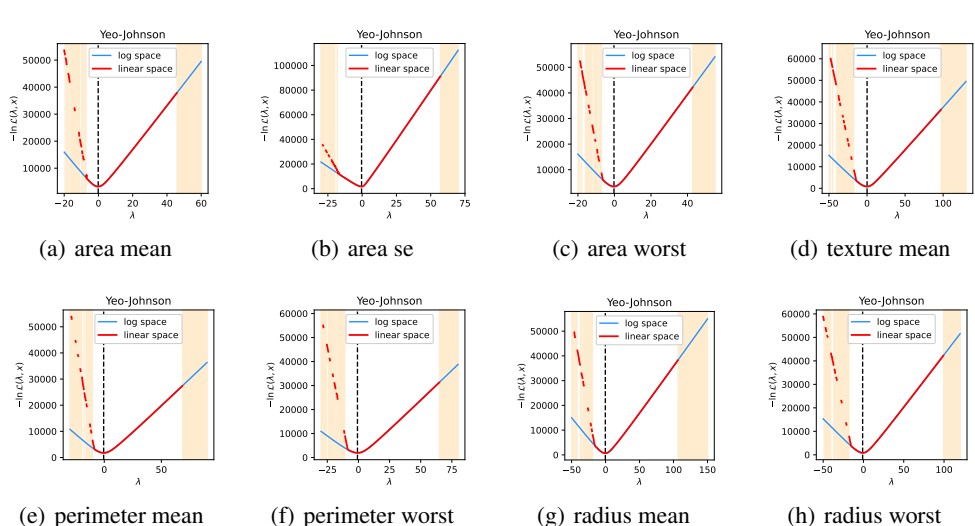

Figure 12: Computation of the negative log-likelihood of Yeo-Johnson as a function of $\lambda$ in log space vs. linear space. The figures use selected features in the Breast Cancer Wisconsin (Diagnostic) dataset.

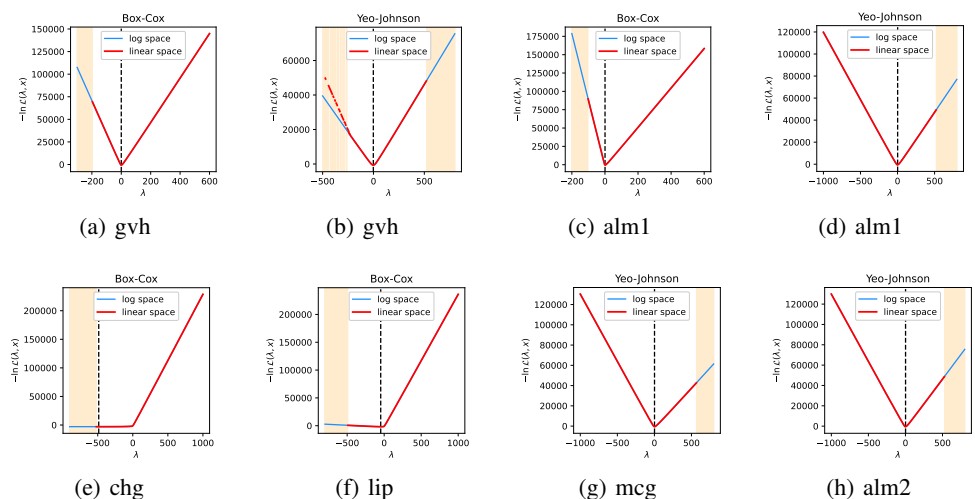

Figure 13: Computation of the negative log-likelihood of Box-Cox and Yeo-Johnson as a function of $\lambda$ in log space vs. linear space. The figures use the rest features in the Ecoli dataset.

