# OpenReview forum: "FedPS: Federated data Preprocessing via aggregated Statistics"
_ICLR.cc/2025/Conference — ICLR 2025 Conference Withdrawn Submission_

### Official Review · Reviewer_onAE · 2024-10-27

**Soundness:** 2
**Presentation:** 2
**Contribution:** 2
**Rating:** 3
**Confidence:** 4

**Summary:**

This paper considers data analytics in federated learning. One application of data analytics is data preprocessing, which requires the estimates of certain data statistics. The submission discusses implementation of data statistics estimating algorithms in federated settings, and performs experiments to showcase the importance of data preprocessing.

**Strengths:**

The problem of data preprocessing/analytics in federated networks is important.

FedPS implements a set of data preprocessing tools, using tools like random sketching.

It considers some numerical issues in one algorithm, and makes the implemented open-sourced as well.

**Weaknesses:**

Algorithmic contributions of this paper are limited. It centers around implementations of previous algorithms. It uses a known Log-Sum-Exp trick to handle numerical instabilities of power transform, as well as clipping the data when their absolute values are too big.

For lots of the analytics tasks (estimating quantiles, estimating heavy hitters, etc) in federated learning, simply adding or merging local statistics may not be optimal. For instance, averaging local medians wouldn’t give us global medians. Various open problems remain.

It is not clear what existing algorithms FedPS uses and why it uses them. There are also differences between one-shot methods and interactive methods to estimate the statistics, which are not discussed in detail.


Minor:
power transform is first discussed without explanation.

**Questions:**

Please see 'weaknesses' for details. It would be good to discuss what is new in FedPS in addition to providing systematic implementations of existing statistics estimators.

---

> ### Author Response · Authors · 2024-11-13
> **Response to Reviewer onAE**
>
> > Algorithmic contributions of this paper are limited. It centers around implementations of previous algorithms. It uses a known Log-Sum-Exp trick to handle numerical instabilities of power transform, as well as clipping the data when their absolute values are too big.
> >
>
> Our primary goal is not to introduce a novel algorithm for training in federated learning but to address a crucial, often-overlooked aspect of the machine learning pipeline—data preprocessing. In this work, we extend various data preprocessing methods to federated settings and analyze their communication overhead. For the numerical issues in the power transform, we proposed a reliable solution based on mathematical properties, providing a robust way to handle instabilities.
>
> > For lots of the analytics tasks (estimating quantiles, estimating heavy hitters, etc) in federated learning, simply adding or merging local statistics may not be optimal. For instance, averaging local medians wouldn’t give us global medians. Various open problems remain.
> >
>
> For estimating quantiles and frequent items, we utilize data sketching techniques, which efficiently summarize data while allowing for the merging of results across clients. For example, when computing medians or quantiles, it does not average local values; rather, it merges sketches to obtain a combined result. Although these estimates are approximate, they provide controllable accuracy. We highly recommend the references in Section 3.3 for more background on the effectiveness of data sketching in these tasks.
>
> > It is not clear what existing algorithms FedPS uses and why it uses them. There are also differences between one-shot methods and interactive methods to estimate the statistics, which are not discussed in detail.
> >
>
> FedPS implements all preprocessing methods detailed in Appendix A, designed to address various preprocessing needs, such as feature scaling, encoding, and missing value imputation. These are fundamental for preparing real-world data before training. Could you clarify what you mean by "interactive methods"? An example would help us better address your question.
>
> > Minor: power transform is first discussed without explanation.
> >
>
> We provide an explanation of the power transform in Section 3.4 of the Preliminaries, which describes how it adjusts data to resemble a Gaussian distribution.
>
> > It would be good to discuss what is new in FedPS in addition to providing systematic implementations of existing statistics estimators.
> >
>
> The integration of data sketching for federated data preprocessing is novel in this context. Additionally, the comprehensive analysis of preprocessing methods and communication costs provides valuable insights for the FL community. Furthermore, our robust power transform addresses limitations found in prior approaches, making FedPS a flexible and practical tool for federated preprocessing.

---

### Official Review · Reviewer_krQd · 2024-11-03

**Soundness:** 2
**Presentation:** 2
**Contribution:** 2
**Rating:** 3
**Confidence:** 4

**Summary:**

The paper considers a scenario in federated learning where data preprocessing needed at the clients. It describes some ways of collaboratively estimating some statistics that are needed in such data preprocessing operations. Some experimental results are shown with simple models with and without the scaling preprocessing.

**Strengths:**

- Data preprocessing is often useful in practice. The consideration of such preprocessing operations in a federated learning scenario can have some practical usefulness.

**Weaknesses:**

- It is not clear what is the main contribution of this paper. It seems to be a straightforward combination of several existing techniques. This is also suggested in the list of main contributions on page 2, which does not include any fundamental technical problem that this paper solves.
- The main paper does not discuss any unique characteristic of federated learning problems, where the privacy of data at clients, often including their statistics, needs to be preserved. There is some discussion on federated algorithms in the appendix, which still focuses on straightforward aspects and misses key federated learning challenges such as privacy preservation, client dropout, etc.
- The experiments use simple models and datasets. IID data distribution is assumed, which ignores challenges related to non-IID data that are mentioned in the motivation

**Questions:**

See weaknesses.

---

> ### Author Response · Authors · 2024-11-13
> **Response to Reviewer krQd**
>
> > It is not clear what is the main contribution of this paper. It seems to be a straightforward combination of several existing techniques. This is also suggested in the list of main contributions on page 2, which does not include any fundamental technical problem that this paper solves.
> >
>
> First, introducing data sketching into federated preprocessing is not straightforward and required detailed analysis of each preprocessing method to identify relevant statistics. Second, our work provides a flexible tool for federated data preprocessing, which we believe is valuable to the FL community. Finally, we propose a robust power transform that we adapt to the federated setting, addressing limitations seen in previous work.
>
> > The main paper does not discuss any unique characteristic of federated learning problems, where the privacy of data at clients, often including their statistics, needs to be preserved. There is some discussion on federated algorithms in the appendix, which still focuses on straightforward aspects and misses key federated learning challenges such as privacy preservation, client dropout, etc.
> >
>
> We address privacy considerations in Sections 7 and 8, where we discuss options for aggregating statistics with privacy-preserving techniques like Secure Aggregation, Secure Multi-Party Computation, and Differential Privacy. For client dropout, the aggregated statistics represent data from a subset of clients. Since aggregation is typically a one-time communication round, dropout does not pose the same challenges as it does for iterative training.
>
> > The experiments use simple models and datasets. IID data distribution is assumed, which ignores challenges related to non-IID data that are mentioned in the motivation.
> >
>
> Non-IID data does not affect the core aggregation of statistics, as we show in Tables 1 and 4. Aggregations such as min/max rely on comparisons, while sum/mean/variance involve summing across clients. The motivation behind our approach is that non-IID data incurs heterogeneous local statistics, which makes aggregated global statistics essential. Additionally, the datasets we use are widely recognized as benchmarks, not “simple” datasets. Finally, simpler models are often preferred in federated settings due to communication constraints, where complex models would incur higher overhead.

---

> > ### Comment · Reviewer_krQd · 2024-11-23
> >
> > Thanks for the response. Some more comments are as follows.
> >
> > > introducing data sketching into federated preprocessing is not straightforward and required detailed analysis of each preprocessing method to identify relevant statistics.
> >
> > I feel that the "detailed analysis of each preprocessing method" is somewhat of a problem here. What I'm missing is a generic algorithm that applies to a broad class of statistics and the analysis of such a generic algorithm in a federated learning setup with possible client dropouts etc.
> >
> > > We address privacy considerations in Sections 7 and 8
> >
> > These are related work and conclusion sections, which do not really talk about the work in the present paper.

---

> > > ### Author Response · Authors · 2024-11-24
> > > **Response**
> > >
> > > > What I'm missing is a generic algorithm that applies to a broad class of statistics
> > >
> > > Data sketching provides a generic framework for computing statistics, the specific algorithms used to construct these sketches often vary depending on the desired statistic (e.g., quantiles, frequent items).

---

### Official Review · Reviewer_toLQ · 2024-11-03

**Soundness:** 2
**Presentation:** 2
**Contribution:** 2
**Rating:** 3
**Confidence:** 3

**Summary:**

The paper addresses the often overlooked aspect of data preprocessing in Federated Learning (FL). The authors introduce FedPS, a comprehensive suite of tools designed to enhance the preprocessing of decentralized data, which is essential for improving the accuracy and performance of machine learning models in federated settings.

The paper highlights the challenges posed by Non-IID data distributions across multiple clients, which complicate traditional preprocessing methods. To tackle these issues, the authors propose a novel Federated Power Transform algorithm that improves numerical stability through log-space computations and constrained optimization, achieving superlinear convergence rates.

Additionally, the paper presents experimental results that demonstrate the effectiveness of the proposed preprocessing techniques, showcasing improvements in model performance compared to existing frameworks. The paper contributes valuable insights and tools for enhancing data preprocessing in federated learning environments, emphasizing its importance for robust model training.

**Strengths:**

1. The paper introduces a robust methodology for federated data preprocessing through the FedPS tool, which leverages aggregated statistics and data sketching techniques.
2. The authors tackle numerical challenges associated with power transforms, which have been a limitation in previous research. By employing log-space computations and constrained optimization, the proposed Federated Power Transform algorithm enhances numerical stability and achieves superlinear convergence rates.
3. The paper is well-structured and well-written, making it easy to understand.
4. The authors provide an open-source implementation of FedPS. The FedPS tool offers a flexible suite of preprocessing options with customizable parameters, allowing users to adapt the preprocessing techniques to their specific needs and data characteristics.

**Weaknesses:**

1. the scope of these experiments is limited in terms of the variety of datasets and models tested. very primitive models are tested, where models like resnet family etc should be experimentally tested with the proposed method.
2. While the paper discusses the necessity of federated data preprocessing in non-IID settings, it does not include detailed experimental results that specifically demonstrate the performance of the proposed methods under non-IID conditions.
3. A detailed privacy analysis of the preprocessing techniques implemented in FedPS is missing which is very important in a federated setting.
4. A more thorough literature review and comparison with federated pre-processing techniques is missing, like Federated One-Hot Encoding, Federated Feature Selection etc.

**Questions:**

1. Can you elaborate on how FedPS compares with the latest advancements in federated data preprocessing, particularly in terms of efficiency and privacy guarantees?
2. How does FedPS perform in non-IID data distributions among clients?
3. Can you provide a detailed privacy analysis of the proposed methods in FedPS, and how do you plan to address this important aspect in the context of federated learning?

---

> ### Author Response · Authors · 2024-11-13
> **Response to Reviewer toLQ**
>
> > The scope of these experiments is limited in terms of the variety of datasets and models tested. very primitive models are tested, where models like resnet family etc should be experimentally tested with the proposed method.
> >
>
> Our experiments aim to show that even a basic preprocessing step, like rescaling each feature to similar ranges, can improve model performance. In practice, preprocessing tasks like feature encoding and missing value imputation are fundamental to any data pipeline. This demonstrates that preprocessing is essential regardless of the model complexity. Our focus on tabular data reflects this, as tabular data involves a broader variety of preprocessing methods. Additionally, simpler models are more suitable for federated learning due to the communication overhead, which is often prohibitive for larger, complex models.
>
> > While the paper discusses the necessity of federated data preprocessing in non-IID settings, it does not include detailed experimental results that specifically demonstrate the performance of the proposed methods under non-IID conditions.
> How does FedPS perform in non-IID data distributions among clients?
> >
>
> All aggregated statistics, as outlined in Table 1 and Table 4, remain unaffected by the non-IID nature of the data. Aggregations like min/max, sum, mean, and variance operate by comparing and summing across clients, so the global statistics are consistent regardless of local data distributions. Our discussion on non-IID settings is intended to highlight the heterogeneity in client data, which further justifies the need for federated aggregation of global statistics.
>
> > A detailed privacy analysis of the preprocessing techniques implemented in FedPS is missing which is very important in a federated setting.
> Can you provide a detailed privacy analysis of the proposed methods in FedPS, and how do you plan to address this important aspect in the context of federated learning?
> >
>
> We agree that privacy is critical and have addressed it in Sections 7 and 8. The aggregation of statistics in FedPS can utilize several privacy-preserving techniques, such as Secure Aggregation, Secure Multi-Party Computation, and Differential Privacy. Each of these approaches offers distinct privacy guarantees, allowing for flexible integration based on the use case.
>
> > A more thorough literature review and comparison with federated pre-processing techniques is missing, like Federated One-Hot Encoding, Federated Feature Selection etc.
> Can you elaborate on how FedPS compares with the latest advancements in federated data preprocessing, particularly in terms of efficiency and privacy guarantees?
> >
>
> We have reviewed the limited literature on federated preprocessing and discussed relevant work in Section 7. Existing studies in this area have not conducted an in-depth exploration of common preprocessing methods, which we summarize in Table 3. While our work does not cover Federated Feature Selection, we provide a comprehensive examination of widely used preprocessing techniques, such as scaling, encoding, and imputation, in the federated context.

---

### Official Review · Reviewer_TmjQ · 2024-11-04

**Soundness:** 3
**Presentation:** 3
**Contribution:** 2
**Rating:** 5
**Confidence:** 3

**Summary:**

The paper addresses the critical yet often overlooked aspect of data preprocessing in Federated Learning (FL), especially in scenarios where data is not identically distributed across clients. The main contribution of the paper is a toolset designed to improve data preprocessing in FL by utilizing combined statistics, data summarization, and federated machine learning techniques. A highlighted contribution is the solution to numerical challenges in power transformations, achieved through computations in logarithmic space and constrained optimization, resulting in a Federated Power Transform algorithm with rapid convergence, inspired by Brent’s method. The authors suggest that FedPS lays a strong groundwork for federated data pre-processing and plan to explore privacy-preserving methods to enhance its privacy features in future work.

**Strengths:**

- The paper provides a solution for a critical yet often overlooked aspect in Federated Learning.
- The code attached to the paper is overall well-written and of good quality, and I agree with the authors claim that FedPS lays a strong groundwork for federated data pre-processing.
- The paper is overall well-written, and the quality of presentation is good.

**Weaknesses:**

The main weakness of the paper is the lack of novelty. The paper is a description of a library for federated data pre-processing. Besides the library itself, the only part of the paper that could be considered as a significant contribution is the proposed algorithm for Federated Power-Transforms, which aims at addressing numerical issues in power transform through logarithmic transformation.

While I overall like this paper and think that it is helpful for the FL community, I don't think that ICLR is the right venue to publish it.

**Questions:**

N/A

---

> ### Author Response · Authors · 2024-11-13
> **Response to Reviewer TmjQ**
>
> > The main weakness of the paper is the lack of novelty.
> >
>
> We understand that novelty can be interpreted in different ways. Our aim in this work was not to develop new algorithms but rather to extend widely used preprocessing methods in the machine learning pipeline to the federated learning (FL) context and to provide a practical library for this purpose. As you noted, one key contribution is the robust federated power transform, which includes our constrained optimization approach and analysis of the Box-Cox function’s properties. Additionally, we contribute by investigating the types of statistics each method requires and analyzing the communication overhead, which is critical in FL settings. For example, for computing quantiles and frequent items—a complex task in federated contexts—we propose the use of data sketching.
>
> > I don't think that ICLR is the right venue to publish it.
> >
>
> We submitted to ICLR’s software libraries track, which we felt aligned with our contributions in this work. However, we’d welcome any suggestions on alternative venues that you feel might be a better fit.

---

### Note · Authors · 2024-11-26

I have read and agree with the venue's withdrawal policy on behalf of myself and my co-authors.